# Workflow for detecting biomedical articles with underlying open and restricted-access datasets

**Anastasiia Iarkaeva**, **Vladislav Nachev, Evgeny Bobrov** *

QUEST Center for Responsible Research, Berlin Institute of Health (BIH) at Charité –Universitätsmedizin Berlin, Berlin, Germany

* evgeny.bobrov@bih-charite.de

## Abstract

To monitor the sharing of research data through repositories is increasingly of interest to institutions and funders, as well as from a meta-research perspective. Automated screening tools exist, but they are based on either narrow or vague definitions of open data. Where manual validation has been performed, it was based on a small article sample. At our biomedical research institution, we developed detailed criteria for such a screening, as well as a workflow which combines an automated and a manual step, and considers both fully open and restricted-access data. We use the results for an internal incentivization scheme, as well as for a monitoring in a dashboard. Here, we describe in detail our screening procedure and its validation, based on automated screening of 11035 biomedical research articles, of which 1381 articles with potential data sharing were subsequently screened manually. The screening results were highly reliable, as witnessed by inter-rater reliability values of $\geq 0.8$ (Krippendorff's alpha) in two different validation samples. We also report the results of the screening, both for our institution and an independent sample from a meta-research study. In the largest of the three samples, the 2021 institutional sample, underlying data had been openly shared for 7.8% of research articles. For an additional 1.0% of articles, restricted-access data had been shared, resulting in 8.3% of articles overall having open and/or restricted-access data. The extraction workflow is then discussed with regard to its applicability in different contexts, limitations, possible variations, and future developments. In summary, we present a comprehensive, validated, semi-automated workflow for the detection of shared research data underlying biomedical article publications.

## Introduction

Monitoring of the scientific output by institutions, funders, and governments has in recent years extended to outputs representative of open science practices. It is of interest to stakeholders to determine e.g. in how far clinical studies are preregistered and timely reported [1, 2]. Other possible criteria are the availability of open code and open protocols, preregistration, and especially open access to research articles, which has received most attention of all open

**Data Availability Statement:** The full dataset underlying this publication and the Numbat extraction forms have been shared on Zenodo (Iarkaeva et al., 2023) and are available under DOI doi.org/10.5281/zenodo.8249758. Free text notes

from extractions, which were not analyzed here, were removed from the uploaded files.

**Funding:** AI was in part funded by the German Federal Ministry of Education and Research (BMBF) and the State of Berlin within the Excellence Strategy of Federal and State Governments through the Berlin University Alliance (https://www.berlin-university-alliance.de/en/), Grant number 312_OpenCall_1. The funder had no role in study design, data collection and analysis, decision to publish, or preparation of the manuscript.

**Competing interests:** The authors have declared that no competing interests exist.

science practices [3]. Open access is so far also the only practice performed by a majority of researchers, at least in biomedical research [1]. Apart from open access, open sharing of research data ('open data') has received most attention, which reflects the calls of governments, funders, journals, scientific societies, and other stakeholders to make data available for transparency and reuse purposes [4–10].

However, to implement a monitoring of open data practices, two components are necessary: First, a definition of what open data exactly is, and second, a workflow to implement this definition for monitoring purposes. With respect to the former, definitions of open data are available [11, 12], and most stakeholders seem to have an overlapping general understanding of what constitutes open data. However, for monitoring purposes an operationalization which is precise and implementable in practice is needed to decide in individual cases, whether open data is present. We have recently published such a set of criteria for open data [13], and described the development of and reasoning behind these criteria in detail in Bobrov et al. [14]. Importantly, this operationalization sets out from the research article, and it is detected, whether for a specific research article underlying data have been shared (openly or under justified restrictions). Here, we present a semi-automated workflow we developed based on above criteria, which allows to put a screening of articles for underlying open and restricted-access datasets into practice.

The workflow presented has been developed in the period 2018–2022 and was introduced to detect publications with underlying open and restricted-access datasets and incentivize these within an institutional performance-oriented funding scheme [15]. Such an institutional incentive for open data sharing which would not require active application by researchers is to our knowledge still unique. At German university medical centers, such incentives are typically based exclusively on article output and/or third-party funding [16]. In addition to the application for incentivization purposes, we have later included open data as a metric in the Charité Dashboard on Responsible Research [1], which displays the prevalence and development over time of different open and responsible research practices at the Charité –Universitätsmedizin Berlin, a large university medical center. So far, there are only a few other dashboards addressing open data practices. In the case of the European Commission Open Science Monitor [17], open data is addressed as a topic, but information is unavailable on the prevalence of actual data sharing. Dashboards based on the OpenAIRE research graph assess the prevalence of open data sharing [18]. However, their approach is very different from ours in that they directly assess datasets available in repositories, while we assess data availability on the article level. Importantly, screenings based on the OpenAIRE research graph and other screenings, including commercial screening services and the HMC Dashboard on Open and FAIR Data in Helmholtz [19], are based on metadata from repositories, and to our knowledge consider any entry to be a dataset as long as this property is indicated in the metadata. However, in our and others' [20] experience this is a simplified assumption, which could easily create skewed numbers of detected datasets. A detailed discussion of these topics is to be found in Bobrov et al. [14]. Notably, the recent French Open Science Monitor–Research Data [21], which is currently in the beta phase, has a similar approach to ours, but uses a different screening tool.

We have recently added a sub-dashboard displaying data reusability (FAIR data) metrics to the QUEST Dashboard [22]. This FAIR Dashboard uses the output of the open data detection workflow presented here, and applies further automated analyses of data 'FAIRness' using the F-UJI tool [23]. The screening for FAIR criteria compliance is, however, not part of the open data screening workflow itself, and is not further discussed here.

In the present article, we describe the steps of the open data detection workflow, focusing on the conceptual level and validation data. A detailed instruction for the performance of the

open data assessment has been published as an open protocol [24]. The automated part of the assessment makes use of the ODDPub tool [25], whose principles and validation have already been described [26]. After briefly describing the assessment procedure, we present results from our own extractions, including a quantitative assessment of three publication samples, as well as measures of interrater reliability and the time invested. We close by discussing the reliability and practical feasibility of the open data detection workflow, and how it could be further improved and adjusted for different contexts.

## Methods

### Extraction of data availability status

The open data detection workflow consists of the following steps. These are described below the enumeration, but **for a detailed description which allows replication of the workflow see Iarkaeva et al. [24]**. Please note that here, diverging from the protocol, we use the term "rater" rather than "extractor".

1. Generation of a list of article DOIs and corresponding full text files

2. Application of the ODDPub algorithm

3. Importing of the ODDPub output into the Numbat screening tool

4. Screening of datasets underlying articles by one or more raters within Numbat

5. Reconciliation of individual assessments, if necessary

6. Cleaning of the resulting data, if necessary

The list of article DOIs in step 1 can often be sourced from an institutional bibliography. In our case, to generate this bibliography, metadata were pulled from the PubMed and Embase databases and consequently harmonized and deduplicated. Metadata include creation date and publication year, which might indicate different years if the given article was published very late in the year and added to one of the databases only in the subsequent year. In addition, in some cases, the publication year entry is added only later to existing article metadata. Thus, as we searched for articles from the Charité for the examined year (here 2021) as soon as they appeared in the databases, there are three publications officially published in 2020 in our sample for 2021. Further inclusion criteria based on the metadata were English language publications classified as research articles. To account for imperfect article type classification in the databases, we also actively excluded publications without abstracts, as well as publications with titles indicating Editorials, Protocols, Comments, Replies, Conference abstracts, Corrigenda, and Obituaries. As a result of this operationalization without complete manual validation we expect that only a very small number of articles fulfilling the intended exclusion criteria still remains in the dataset. Furthermore, as we aimed to prevent the unnecessary download of articles matching our exclusion criteria, the number of articles not screened was a mixture of publications that we excluded a priori and publications that we could not access. Reasons for lack of access were lacking institutional rights or barriers against automated download placed by the journals (sometimes also for open access articles). Some publications were discovered to have been misclassified in the databases only after download, and thus the filtering was adjusted in an iterative process.

In step 2, the algorithm ODDPub [25] was applied. This text mining tool implemented in R programming language screens article full texts (in PDF format) for statements indicative of openly shared datasets. The methodology and validation have been described in detail by Riedel et al. [26]. Please note the following restrictions: (i) ODDPub has been developed and

validated for the biomedical research domain. In particular, the repository search is facilitated by a list of repository names, and repositories from other domains might be missed. (ii) ODD-Pub's main development took place in 2019, and new developments regarding infrastructures and data referencing practices might decrease the detection performance. An update of ODD-Pub is currently underway and will e.g. consider the increase in the prevalence of data availability statements. (iii) Access to the article full-texts is a prerequisite and thus, depending on institutional access rights, a substantial portion of articles might be excluded from analysis. (iv) The extraction workflow described here includes not only the detection of fully open datasets, but also datasets shared with restrictions as well as data reuse cases. However, ODDPub has only been validated for the detection of fully open datasets, and thus, the false negative rates in the other cases mentioned could be higher. (v) The approach taken by ODDPub necessarily means that datasets shared independently of articles are not detected and not considered further.

In step 4, a template implemented in the Numbat Systematic Review Manager [27], written in the programming language PHP, was used to check in a stepwise fashion compliance with a set of open data criteria. The rating started with confirming, whether the statement detected by ODDPub was actually a clear reference to a shared dataset (or a nearby section in the article was). Subsequently, it was checked, whether the dataset could be found, was shared in a repository, could be accessed, was in a machine-readable data format, and consisted of raw data which would presumably allow analytical replication of at least some analyses. In addition, it was checked whether there was overlap between the creators of the dataset and the authors of the article in question. The individual steps are also shown in the flowcharts in Fig 1 as well as S1 and S2 Figs. This procedure was repeated for datasets deposited in different repositories, where applicable.

We considered datasets to be open or restricted-access data if they complied with the following criteria:

1. The shared output is research data

2. Data have been generated or collected by author(s) of the article

3. Data availability is clearly indicated in the article in question

4. Data can be found

5. Data can be accessed

6. Data are raw data in a reusable format

7. Or, as an alternative to (5) and (6): data are personal data shared under restrictions, with a clearly defined access procedure

These criteria are already more detailed than most definitions for available data. Nevertheless, they do still not allow to determine the data availability status in a useful and reproducible way. For each of these criteria, a more precise definition and operationalization is needed. Due to the occurrence of diverse types of data deposit, including rare and sometimes idiosyncratic borderline cases, this operationalization becomes quite extensive, and thus goes beyond the scope of this article. The exact definition of criteria and a rationale for our choices are provided in detail in Bobrov et al. [14].

For each article, information could be extracted for one or more datasets. This was implemented using "sub-extraction" forms, nested within the Numbat article-level extraction form. In step 5, the screening results from individual raters were reconciled (see below) for publications screened by more than one rater. More than one rater was involved for the validation

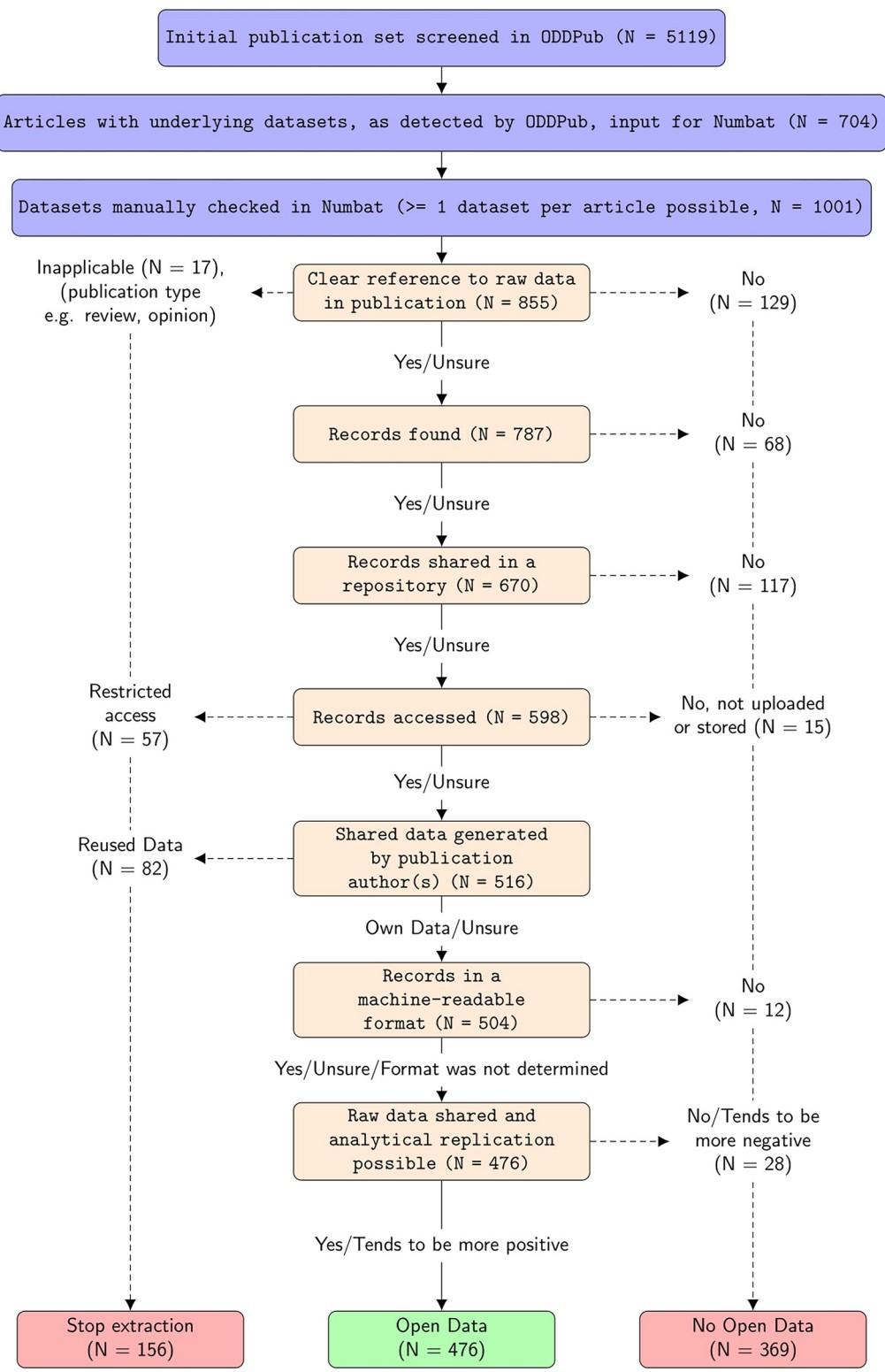

**Fig 1. Flowchart of screening steps to determine the data availability status of articles published in 2021 by researchers from the Charité –Universitätsmedizin Berlin.** Numbers in beige boxes indicate the number of articles screened at the respective stage which complied with the criterion in question. The number referring to "stop extraction" includes articles with restricted-access datasets. These were not "open data" in the full sense, but counted as such for our incentivization scheme. The labelling "unsure" is for illustrative purposes only. Where compliance with a

criterion was assessed as "unsure" by an individual rater, the extraction continued, but the overall reconciled assessment was always either "yes" or "no".

samples, used to calculate interrater reliability (IRR), as well as for datasets where the first rater was unsure. Lastly, in step 6, the dataset required cleaning, depending on the use case. E.g., in our case, for the allocation of an incentive for data sharing, we aggregated the assessment on the article level, and detected, whether for a given article an open (or restricted-access) dataset was available in **at least one** repository. The presence of further datasets not compliant with open data criteria does not preclude the allocation of the incentive.

The reconciliation function in Numbat was used to compare extractions on the dataset (i.e., "sub-extraction") level from different raters and consolidate them into a final overall assessment. This function allowed to transfer extracted information from any rater into the final rating form, as well as manually changing the answers in the final form. The reconciliation form for sub-extractions consisted of columns, with each column representing a different rater. Reconciliation was only possible for sub-extractions that had been completed by more than one rater. Within the list of sub-extractions, it was possible to move or swap them up or down until similar extractions for each rater were placed side by side for validation. To begin reconciliation, it was considered sufficient that datasets originated from the same repository (or other deposit location). Typically, all raters had extracted the exact same dataset, but this was not a prerequisite for reconciliation. If, for a particular repository, the raters assessed different datasets from the same deposit location, but all raters arrived at the same result, any one of the datasets could be transferred to the final form. However, if different datasets from the same deposit location resulted in differing assessments of open data status, an overall assessment of the respective case as 'open data' would either require unanimous agreement among all raters in a discussion session, or a dataset so far only covered by one rater would be assigned to another rater, ensuring that two assessments were available for further discussion. The raters who performed most extractions had expertise in data management/biology and information science/linguistics, respectively. Further extractions were performed by raters with similarly diverse backgrounds, most of them unrelated to medicine. The experience level was also diverse, from student assistants to experienced staff scientists.

Disagreements with regard to the final assessment, i.e., whether the dataset was to be considered "open data" or "not open data" were discussed by two raters until a consensus was reached. In very rare cases a third rater was involved to justify his or her assessment and help build an overall consensus. The answer "Unsure, discussion needed" by only one rater did not necessarily lead to discussion. The rater performing the reconciliation had at his or her disposal all arguments which other raters documented with their choices. If the rater performing the reconciliation, based on these arguments, came to an overall conclusion which he or she was confident of, no further discussion took place. On the other hand, if the rater was unsure of the final assessment, this article was discussed until an agreement was reached. The assessment 'unsure' in any other question than the final assessment was only discussed if more than one rater gave this assessment or if the disagreement was at the root of the disagreement in the final assessment. If a dataset was missed by one rater, the procedure depended on whether the other extraction(s) of this dataset were clear-cut and consistent. In rare cases, the rater who missed a dataset redid it as a basis for reconciliation. The details of the reconciliation procedure are largely intuitive but also intricate. Further information can be found in the published protocol [24].

## Calculation of Inter-rater reliability (IRR)

We calculated the IRR as Krippendorff's alpha [28] based on the overall assessment of the dataset accessibility status, i.e., on the final outcome of the extraction. The six possible outcomes were: (1) yes (= open data), (2) no (= not open data or no dataset available), (3) unsure, (4) restricted-access dataset, (5) inapplicable (= no datasets were generated in this research), and (6) data reuse. We did not calculate the agreement between raters for individual assessment steps, but this would probably have been lower than the agreement for the overall assessment.

IRR was calculated in two samples: a sample of 100 publications extracted by two raters, and a sample of 20 publications (overlapping with the larger sample) extracted by three raters. The validation samples were drawn randomly from the overall set of articles. The Numbat extraction tool allows raters to extract data independently from each other based on the same articles and datasets. For each article one or more datasets could be extracted. Thus, IRR of data availability status was calculated in two ways–on the article level and individually for each dataset detected in an article. In the article-based assessment, for those cases where information was extracted for more than one dataset, we first determined for each rater individually whether the article had at least one open data case and then compared the results between raters. Thus, this comparison was based on 20 and 100 assessments, respectively. In the dataset-based assessment, IRR was calculated based on all datasets extracted by all raters, and thus based on >20 and >100 cases, respectively, as several datasets per article could be considered. Datasets considered by some raters, but not others, were included in the calculation. We had originally calculated IRR based only on those datasets extracted by all raters. This number is more indicative of the agreement regarding specific datasets, but less indicative of the reliability of the overall result. As the latter is of primary importance, we do not report these original IRR values anymore, which were, however, similar.

We quantified the occurrence of such disagreements between raters regarding the question of how many datasets had been extracted for a specific article. For two raters, such a disagreement occurred 20 times for 100 publications. The disagreement observed is in large part due to inconsistencies regarding the screening of so-called "source data" (supplements shared on the publisher website, containing the data which underlie article figures). These and other cases of disagreement were typically not considered open datasets, and only very rarely influenced the overall open data assessment on the article level. In addition to missing a potential dataset in a specific deposit location, it was also possible that raters extracted different datasets for the same repository. Such differences were not regarded further, and these extractions were considered as equivalent to each other in the IRR calculation. Here, we worked under the assumption that the dataset characteristics for multiple datasets deposited in the same repository for the same article are similar. This is in line with our qualitative observations, but we did not confirm that quantitatively.

Please note that there is potential for methodological flexibility in the IRR calculation, and we did not preregister our calculation. Variations would be possible, which would change the exact numbers. Also, the IRR calculated here focuses on the data availability status on the level of datasets and articles. It does not, however, focus on the eligibility for the open data incentive. Here, the IRR would have been higher, as "yes" and "restricted-access" would both be pooled. What is more, the IRR would have been substantially higher if a high precision on the level of individual assessments had been the primary focus. However, we rather focused on the precision of the overall assessment. To avoid erroneous overall assessments, we chose the "unsure" category liberally and thus channeled more articles into the reconciliation procedure. This came at the cost of decreased precision of individual assessments, as well as of additional time spent on reconciliations. For 146 datasets screened by two raters (derived from 100

randomly assigned article-level extractions), only in 13 cases was there a fundamental disagreement between raters such that final assessments were in mutually exclusive categories ("unsure" not being in disagreement with any specific category). Another type of disagreement occurred when one rater chose one of the final assessment categories and another chose "unsure". This occurred five times in total. In three cases there was a "pseudo-agreement" when both raters chose "unsure". The final assessments in cases of disagreement were made during a discussion to reach an agreement.

Krippendorff's alpha is a universal metric for reliability of assessments between raters, as it can be applied to any scale of measurement, any number of raters, and works similarly well with and without missing data [28, 29]. Alpha can take values between 0 and 1, which represent the absence of any agreement and perfect agreement between raters, respectively [28]. We used the package DescTools in R [30], containing the function KrippAlpha for calculating the IRR values [31]. Moreover, the results could be replicated with the independent R package irr [32, 33].

## Calculation of the average duration of extractions

The time spent on extractions in Numbat was determined by starting a timer at the beginning of extractions (i.e., when a new browser window of an extraction is opened). It is particularly important to note that the timer in the extraction form and the timer in the reconciliation form are distinct. Here, we looked only at the time spent on extraction, not on the subsequent reconciliation of results between raters. Thus, it cannot be used to directly extrapolate the overall time needed to assemble a dataset with a given number of ratings. Furthermore, the timer was only stopped when the overall extraction was finished (by pressing the "complete" button). Thus, extraction duration was recorded on extraction level (article), not the sub-extraction level (dataset). As a consequence, the durations indicated for extractions on the dataset level are derived by dividing the overall duration by the number of datasets extracted. This calculation was performed for the extractions of all raters combined.

Importantly, the time spent on extractions is only a very rough estimate. This is due to multiple factors, one of which the difference in the number of datasets rated per article by different raters. This confounder is not present in the analysis of extractions on dataset level. Several other factors, however, complicate the interpretation of the durations on both article and dataset levels:

i.  extraction of information before opening the extraction window

ii.  different level of detail with which text around the statements detected by ODDPub was inspected

iii.  time spent researching practices regarding repositories and data referencing

iv.  different level of detail regarding the documentation of the decision taken, including unusual cases and questions arising during the process

v.  distractions unrelated to the extractions

vi.  corrections made to the extractions after clicking the "complete" button

In some cases, the extraction times documented were clearly unrealistically long. This might have been caused by either not clicking the "complete" button in the end, or by not stopping during long breaks taken while extracting. For the Charité dataset, overall 872 articles were extracted by three raters (704 unique articles), resulting in 1236 extracted datasets. For detecting outliers with respect to extraction duration, we calculated the interquartile range

(IQR) for the duration of extraction (on the article level, as the time stamps were only logged on that level; see above). This range was then multiplied by 1.5 and subtracted from the first quantile for the lower limit (Q1–1.5*IQR), or added to the third quantile for the upper limit (Q3 + 1.5*IQR), respectively. This way, we excluded 39 out of 872 cases.

For the overall average duration on the article level, individual durations were averaged over all raters. Thus, raters who screened more publications had a larger impact on the average duration. As the dataset-level average duration was derived from the article-level duration, the same also applied here.

## Calculation of the performance of ODDPub

In a previous report on ODDPub [26], a validation set of 792 publications was used for algorithm development, and these publications were all screened manually to serve as a baseline reference or "ground truth". Thus, for this set, all the automatedly detected and manually confirmed values of both positive and negative detections were available to quantify performance captured by metrics like sensitivity, specificity, and F1 score. In contrast to that, the detection metrics reported here are derived exclusively from the articles detected as 'open data articles' by ODDPub. Thus, false negative and true negative values cannot be calculated. However, as a plausibility check for the current performance of ODDPub, we manually screened a sample of 100 articles from 2021 which were not flagged as open data by ODDPub. In this sample, ODDPub missed one restricted-access article, which indicates a lower false-negative rate as compared to the original validation of ODDPub (3.4%). This sample should primarily be seen as supportive evidence for a continued reliable detection by ODDPub. However, if it were indicative of an improved performance, this would be in line with the fact that most cases (17/24) missed by ODDPub in the original manual validation were supplementary materials, which are now not within our definition of available data.

All data underlying this publication, as well as the extraction forms used, have been openly shared on Zenodo [34]. See the data availability statement for details.

## Results

### Data availability for Charité publications

For the 2021 screening (i.e., articles published in 2021) we screened an overall corpus of 6159 articles by Charité researchers. These were detected by affiliation search in the databases Pubmed and Embase, with subsequent deduplication, as described in detail in Riedel et al. [26]. Of these, 5119 article full texts were screened with ODDPub, resulting in 704 "hits" ("open data automated"). The DOIs and detected data statements for these articles were imported into Numbat and screened manually. For 55 articles, the overall assessment of open data status by at least one rater was "unsure". These articles were reconciled to yield an overall consensus assessment. For the 704 articles manually screened, 1001 potential datasets were manually extracted, with only one dataset extracted per repository per article. Of these datasets, 476 were confirmed to be open datasets by our definition. On the article level, the open data status was confirmed by manual check for 397 articles, i.e., at least one dataset had been shared openly. This corresponded to 56.4% of articles detected by ODDPub (precision), and 7.8% of the overall sample of articles screened. In addition, we found 57 datasets with restricted access (i.e., data shared with justified restrictions, where there was a standardized way to access the data). These were related to 53 unique articles with one or more restricted-access datasets (1.0% of the overall screened sample of articles). Dataset access types could be overlapping for one article, such that an article had data shared both openly and under restrictions. Overall, 424 (= 8.3%) of articles in the Charité 2021 dataset had open and/or restricted datasets shared.

This confirms a trend towards more available datasets since the screening was started. Between 2016 and 2021, the fraction of Charité articles with fully open datasets rose from 3.0% to 7.8%. The increase in absolute numbers is much higher, but is offset by an increase in the number of articles by >50% between 2016 and 2021. Please note that supplements deposited on publisher websites are now excluded from our analysis and the Charité dashboard [1] for all years, while they had been included in numbers presented earlier. Nevertheless, the numbers reported here slightly differ from the dashboard, which is due to different time points at which the literature lists were generated, as well as subtly different literature search algorithms.

On the level of datasets (n = 1001 checked), the most common reason for a potential dataset not to be considered "open data" (fully open or under restricted access) were cases where manual screening did not confirm a clear reference to a shared dataset in the article text (n = 129). It was also commonly observed that the deposit of the specific dataset in question had not been made in repositories (n = 117)–most commonly these were supplementary materials on publisher websites. Another common case of exclusion were cases where a dataset was referenced, but could not be found (n = 68). This was e.g. the case where only the repository name was given, but not the accession code, or where the dataset was linked in the article, but was not in itself findable in the repository, as defined in our criteria. In contrast, we only rarely observed that datasets had been shared in formats which were not machine-readable (n = 12). For a detailed discussion of the criteria and different borderline cases see [14]. We also extracted cases of data reuse, which was observed for 82 (= 1.6%) of articles. An overview of all decision steps and the corresponding numbers along the extraction pipeline are shown in Fig 1. Please note that in the assessment of whether raw data had been shared, we implemented a graded assessment in Numbat (yes/tends to be more positive/tends to be more negative/no). However, we did not use this information further, and treated it as a dichotomous yes/no decision.

According to Riedel et al. [26], the negative predictive value of ODDPub is 0.97. Thus, with 4415 out of 5119 articles being negative, it would be expected that we missed 132 cases of open data. However, as supplementary materials are most commonly missed by ODDPub, and we do not consider them open data anymore, unless they are deposited in a repository, we expect a substantially lower number of false negatives.

For articles published in 2021, we analyzed the repositories in which datasets were deposited. By far the most often used repository was Gene Expression Omnibus (GEO), with 106 datasets. This was in line with the overall observation that the disciplinary repositories which are most popular at our institution are from the fields of genomics and proteomics. The top seven most used disciplinary repositories were all from these two fields. Many datasets were also deposited in general-purpose repositories, with Figshare (n = 63), Open Science Framework (n = 46), and Zenodo (n = 39) being the most popular, and overall the second to fourth-most popular behind GEO. Institutional repositories are not commonly used, and none was used more than twice. The distribution of datasets over repositories is displayed in S1 File.

On the level of articles, we analyzed the relationship of dataset availability and article access status. The probability of data underlying an article being available (open or restricted) was approximately twice as high (9.5%) for open access articles as it was for paywalled articles (4.8%). In the sample screened, 72.5% of articles by Charité researchers were openly accessible, i.e., published as green and/or gold open access articles. Importantly, however, the sample screened does not cover all articles published by our institution in 2021. 83.1% of articles were screened with ODDPub, while the remainder was not screened for lack of institutional access, technical reasons, and/or as the respective article type did not produce data (e.g. corrigenda, obituaries; see exclusion criteria in Methods). Thus, the shares of available datasets and even more so of open access articles are approximations to the full institutional output.

To summarize, we would like to reiterate the different samples considered: 6159 articles were published in 2021 overall, of which 5119 were screened with ODDPub. Of these, 704 had open or restricted data according to ODDPub. For these, we manually screened 1001 putative datasets (only one per given repository) and confirmed 476 open datasets, as well as 57 restricted datasets, pertaining to 424 articles. Based on true positives and false positives at the article level, we obtained a precision (positive predictive value) of 0.602. Thus, article publications with an open data statement detected by ODDPub have a probability of 60.2% to be manually confirmed as having at least one underlying available dataset (open or with restricted access). 397 out of 704 screened articles had at least one fully open dataset, resulting in a precision on the article level of 0.564 if considering only fully open data.

We began determining the open data status of articles by Charité researchers in 2018 for articles from 2015 and 2017, and used the experience gained to further improve it in subsequent years. In 2021, we then assembled the complete extraction workflow including ODDPub and Numbat and screened the publications from 2020 using this workflow. After that, the extraction process was optimized further. As the open data criteria and the Numbat extraction form differ between the extractions for 2020 and for 2021, we focused in the results on the numbers from the 2021 extraction. The results for 2020, which are consistent with the other datasets screened, are documented in S1 Fig. The main difference in extraction between these two years was that for 2020, only one dataset per article could be extracted, while in 2021, several datasets could be extracted per article (range 1 to 5) using the sub-extraction function of Numbat.

A higher number of open data instances was detected by ODDPub in 2021 (704 cases) as compared to 2020 (498 cases). However, the share of these statements relative to the total number of screened articles was similar in both years– 12.8% in 2020 and 13.8% in 2021. Similarly, the percentage of open data manually confirmed, out of the automatedly detected sample, was similar– 53.4% in 2020 and 56.4% in 2021. There was an increase in the reuse of data (0.8% vs. 1.7%), as well as in data with restricted access (0.7% vs. 1.0%). However, the occurrence of these two categories is not fully comparable between years due to changes in the order of questions (see Fig 1 and S1 Fig for more information).

## Data availability for Dutch university medical centers

In an independent analysis, Haven et al. [35] screened a subset of 2042 articles published by biomedical researchers at four university medical centers (UMCs) in the Netherlands. They found that out of these articles, there were 179 articles with shared datasets detected by ODD-Pub, and 191 datasets were manually checked in Numbat. Of these, manual screening confirmed 100 articles (= 4.9% of all articles) to be open data and 6 (= 0.3%) to be restricted-access data. Overall, on the article level, by the same criteria as applied to the Charité dataset, 106 publications (= 5.2%) had shared at least one open or restricted-access dataset (ODDPub precision = 0.592). For the Dutch UMC dataset, S2 Fig shows the corresponding numbers for the extraction pipeline.

## Interrater reliability (IRR)

From the 704 articles detected as potentially open data by ODDPub in the Charité 2021 dataset, two subsets were screened by multiple raters to determine the IRR of the overall open data assessment on the article level (i.e., whether for a given article at least one dataset had been shared, either open or restricted). For a subset of 100 randomly assigned articles, assessed by two experienced raters, the IRR was 0.901 on the article level. For a subset of 20 articles, assessed additionally by a third, less experienced rater, the IRR was 0.826. We also calculated

the IRR on a set of 146 datasets, derived from the aforementioned 100 articles. This dataset-level IRR was 0.810 for two raters, and 0.793 for three raters.

## Average duration of extraction

For three raters in the Charité dataset screening, the average extraction time per article after removal of outliers (based on 833 extractions overall) was 4 min 28 sec and the median was 3 min 44 sec. On the dataset level, the average extraction time was 2 min 41 sec and the median 2 min 17 sec.

For the six raters–extracting data for either the Charité or the Dutch UMC datasets—the average extraction time per article after removal of outliers (calculated in a combined dataset for both samples, resulting in 1199 extractions overall) was 4 min 39 sec. The overall average extraction time on the dataset level was 3 min 14 sec.

## Discussion

### Main findings

In this study, we present a workflow which allows to detect whether for research articles underlying data have been shared. This workflow is the first which integrates an automated screening step [26] and a manual confirmation step, thus ensuring a very high degree of reliability (see [24] for the workflow protocol). Such a high degree of reliability might be desired especially where the screening takes place for the purpose of incentivization, but is also an asset to obtain reliable results for monitoring or meta-research. Importantly, this workflow draws on a detailed set of criteria for open as well as restricted-access data [14]. In combination, very reliable extraction results can be obtained.

By applying the extraction workflow to two use cases, we show that it is feasible and reliable in two different contexts. First, we applied it to publications of researchers at the Charité –Universitätsmedizin Berlin, a large German university medical center with >6000 publications per year. Second, we draw on a separate dataset generated for a meta-research study on the effects of supervision on open science practices [35]. Here, data were extracted for publications by researchers at four Dutch university medical centers. For the Charité dataset, the purpose was both to incentivize researchers who shared data [15, 36] and to display the findings in a dashboard [1]. For the Dutch UMC dataset, the purpose was to correlate data sharing practices of supervisors and supervisees.

For the Charité 2021 publications sample we find that 7.8% of articles had underlying openly available data, and 1.0% of articles had shared restricted-access datasets. This resulted in overall 8.3% of articles for which at least one dataset was available. As for a given article both open and restricted-access datasets could be available, the overall percentage is not the sum of both availability types (open and restricted). It is important to note that restricted access only applied to data shared in a repository and with a defined access procedure (for details on how we defined restricted access see [14]). For the Dutch UMC sample, we observed 4.9% of articles with open data, 0.3% with restricted-access data, and 5.2% in combination. The performance of ODDPub regarding false negatives was very similar for both samples. For the Charité 2021 sample, 60.2% of articles flagged by ODDPub were manually confirmed as referring to available data, while for the Dutch UMC sample it was 59.2%.

The numbers we obtained for the Charité dataset are well aligned with numbers so far reported. In the largest analysis so far conducted, Serghiou et al. [37] applied ODDPub to 2.75 million open access articles in PubMed Central (from 1959 till 2020, with the largest numbers in most recent years). They identified mentions of data sharing in 8.9% of cases. Adjusting for discrepancies between predicted and true estimates observed in the validation set, they

estimated that data sharing information was mentioned in 14.5% (95% CI, 11.0% to 18.8%) of articles. This number is very close to the 13.8% we obtained in the automated screening for the Charité 2021 dataset, and suggests, that ODDPub is well suited to detect datasets shared in the wider biomedical research domain. Our findings are also in line with studies based on smaller samples, e.g. by Strcic et al. [38], who screened 897 preprints on COVID-19 and observed 13.8% of preprints with data sharing. A recent analysis of datasets referenced in articles published by PLOS included a comparator dataset of 6000 open access publications randomly drawn from Pubmed [39]. Here, two different estimates of data availability were calculated, of which the more conservative one (requiring deposit in repositories) is relatively close to the criteria we applied. In this case, an availability of datasets was observed for 14% of articles screened, again very close to our observation.

For the Dutch UMC dataset, we observed a lower data sharing rate of 8.8% as detected by ODDPub, resulting in 5.2% after manual confirmation. This dataset might be less representative, which might be amplified by its smaller size (179 articles detected by ODDPub). Different explanations regarding the specificities of Dutch UMCs and the PhD projects conducted there are possible, which would warrant further investigation. One possible reason could be different specializations within biomedical research. Due to the different data sharing rates by research field, which are strongly suggested by the repositories used by Charité researchers [22], such differences in research focus could also strongly influence data sharing rates. In line with that, Rowhani-Farid et al. [40] observed that open data was only available for 4% of articles in BMJ Open. Given the clinical scope of the journal, such a low rate might be due to difficulties of sharing personal data, and it might have been the case that in the Dutch UMC sample, there was similarly a stronger focus on clinical research, where data protection hampers data sharing.

The positive predictive values are comparable for the Dutch UMC dataset (0.59) and for the Charité 2021 dataset (0.56). They are both somewhat lower than the 0.73 reported by Riedel et al. [26] for another Charité dataset. The main reason seems to be an increase in the presence of data availability statements in research articles, which are sometimes, in the presence of other expressions, flagged by ODDPub as cases of open data. The updated version of ODDPub we are currently working on will address this, and we thus expect the positive predictive value to rise again.

As shown in the flow diagrams, many more articles were excluded at certain extraction steps than others. In the Charité 2021 dataset (Fig 1), the most common requirement not fulfilled was the first, requiring that there was a clear reference to the dataset in the publication. These are those cases which are most directly attributable to erroneous detection by the algorithm. In contrast, exclusion of datasets at later stages is increasingly based on our specific criteria, and could be assessed differently using different criteria and definitions, as e.g. in our requirement for data to be raw data. Importantly, extraction stopped, once a criterion was not complied with. Thus, the number of datasets tested is diminishing with every step. Given this decrease in number along the extraction pipeline and the specific subset of datasets which did not reach later extraction steps, it is not possible to directly compare the number of datasets excluded at different steps. It is also noteworthy that across all three samples, only 0.26% of articles were manually detected as not creating data (e.g. opinion, comment). Thus, the pre-screening we performed to select original research articles worked very reliably.

## Scope and applicability

We used the screening at the Charité to incentivize researchers at our institution by assigning performance-oriented funding to publications with shared data [15]. In addition, results were

also used to monitor institutional data sharing practices in the larger context of a responsible metrics dashboard [1]. These are the use cases for which the extraction workflow was developed, and thus it is particularly useful here. Most importantly, we focused on a high fidelity of results. Based on detailed criteria and with a defined multiple-step checking procedure, we make sure that we do not assign the incentive erroneously. To make sure that, at the same time, only a relatively small number of articles with shared datasets is missed, we designed the ODDPub screening to be relatively permissive. In combination, we obtain a comprehensive sample of articles for which data sharing has been manually confirmed, and with a relatively low number of false negatives, although a substantial number still remains, given the imbalance in detection (14% of articles flagged by ODDPub as 'open data', but 86% not flagged). This resource-intensive procedure (see below for a more detailed discussion) is most useful when a high quality of data is desired. However, this must not only be in the context of institutional incentivization. The results by Haven et al. [35] show a strong effect of manual screening. In their article, they calculated the correlation between supervisor and supervisee data sharing rates. While the automated output of ODDPub resulted in an odds ratio of 2.09, after manual checking, the odds ratio increased to 3.74. This suggests that the manual checking resulted in a set of articles which more closely represented actual (or relevant) data sharing practice. Thus, the screening workflow described here might also be useful for focused meta-research projects based on relatively small samples.

Furthermore, the screening workflow presented here could also be used by research funders to assess compliance with data sharing requirements. Given the relatively limited number of grants awarded by most funders, such a screening seems to be feasible. In addition to large institutions as ours, of course the same procedure is equally applicable to research consortia, institutes, or any other academic structure smaller in size than a university.

The Numbat extraction also yields a list of dataset identifiers, which can be used for further analyses. In our case, it was used to apply the F-UJI tool [23] to determine the FAIR criteria compliance of Charité datasets, and to construct a specific FAIR Data Dashboard [22]. Such a dashboard, which will be described in detail in a separate publication, constitutes an additional basis for institutional data sharing strategies, as well as input for education and consulting on data management. Such datasets can then also be reused by repository providers, which might not be interested in detecting data sharing rates, but might use information on how FAIR their repository is, especially as compared to others.

## Requirements and resources

The time one rater needed to screen the availability of data in a research article was on average approximately 4.5 minutes. This duration might seem quite low, as it would result in approximately 53 hours to screen the complete Charité publication set. However, the overall time investment is in fact much higher. The most time-consuming step, once the ODDPub screening process has been established, and apart from the initial screening by raters individually, is the clarification of unsure cases with further experts. This can take the form of either assigning unsure cases to another rater, with subsequent reconciliation, or of a clarification meeting with a subsequent update to the original extraction form. Either way, to compare disagreeing assessments, weigh the rationales behind them, and ultimately arrive at a consolidated assessment takes considerable time. Other steps like the assignment of articles or the documentation of the rationale behind certain assessments also take time, and of course the extraction constitutes quite monotonous work, which cannot be regularly performed several hours at a time and thus needs to be stretched out. The learning curve might contribute to extractions becoming faster over time, but we did not investigate this quantitatively. Average extraction times also

vary considerably between raters. Overall, it is quite difficult to attach even a range of expected time investments, given the differences between extraction settings, e.g. when it comes to the question of whether a subset of articles is to be routinely screened by more than one rater for quality assurance purposes. With aforementions cautions, it can be said that in our experience, to determine the open data status for Charité publications in a thorough way, we needed **very approximately** a full-time work week (39h) per 100 publications, if fully focusing on this task. This can serve as a minimal duration when planning a similar extraction (but see below for a discussion of ways to reduce this duration). Also, the duration does not scale completely linearly, as some sub-tasks like the article screening with ODDPub and the assignment of articles to raters in the best of cases need to be done only once.

For applicability, it is furthermore important to consider that there is of course a learning curve to be taken. This is difficult to quantify, but as a rule of thumb, the extraction of 50 articles needs to be closely monitored or ideally fully validated by an experienced rater. The raters of the samples here presented had different study backgrounds and different levels of experience with scientific work. Although some experience with biomedical research definitely helps, in our experience it is well possible for advanced students also from other research fields to become very reliable and fast raters over time. Always, but especially for raters with little experience, it is recommendable to err on the side of caution and have unsure cases assigned to a second rater.

The overall time investment also strongly depends on the use case. For certain purposes, it can be appropriate to not have any manual screening at all, but if a manual screening is desired, it can be performed more or less thoroughly. In our case, as the outcome is used to distribute research funding, we were very careful both regarding individual extractions, and the overall reconciled assessments, as confirmed by high inter-rater reliability values. For other uses cases, time could be substantially cut by having less or no overlap between raters, and spending less time on complicated cases. In addition, it would be possible to classify datasets from certain repositories as open data right away, not finishing the extraction form. This would, however, need to be added to the final open data status table manually or from another source. Furthermore, it can be sufficient for some use cases to extract information only until one open or restricted-access dataset has been found, and disregard any further datasets mentioned.

Further resources involved computational resources and access to articles. As described in the protocol [24], both ODDPub and Numbat can be installed and used locally, even on a regular laptop. However, it is recommended to run Numbat as a server, which enables central storage of information and performing reconciliations among multiple users on different devices. A noteworthy current limitation in the workflow is that not all publishers provide application programming interfaces (APIs) for text and data mining of full-text publications. Therefore, usually only a subset (typically between 70% and 90%) of the publication record is available for automated download and screening, and this varies additionally with institutional subscriptions and open access status of the articles.

## Validity and limitations

The workflow described here is the first of such detail, and in our opinion provides a high degree of validity and reproducibility. For two experienced raters, the interrater reliability (IRR) on the article and dataset levels was 0.901 and 0.810 respectively. An IRR of >0.8 has been described by Krippendorff [41] as "reliable". Even for a much smaller sample including a third, less experienced rater, there was high agreement (article-level IRR = 0.793). Thus, already the individual assessment is of high reproducibility, and for an experienced rater it

can, for many purposes, be legitimate to not include a second rater at all. This is also supported by the fact that for 100 articles screened by two experienced raters, there was never a case which would have led to altogether misclassifying an article as "no data available" had the decision relied on either rater alone.

The inclusion of a less experienced rater might have contributed to the slightly lower IRR value for the smaller set of articles assessed by three raters. Importantly, maximizing the IRR was not a goal of our procedure. The extraction procedure was rather set up to increase the reliability of overall assessments, at the expense of a lower reliability of individual assessments. The "unsure" category was thus applied in a deliberately liberal way to make sure that in cases of doubt, a second rater would assess the case. For this reason, the IRR should also not be mistaken for the fidelity of the overall extraction. Due to the application of the four-eye principle with subsequent reconciliation wherever there was doubt, the overall fidelity of the extraction nears 100%. This at least insofar as 100% can be conceptualized here, where there is no natural gold standard. In this case, we would take a fidelity of 100% to be reached by a majority vote of an infinite number of raters.

The following measures contribute to the high reliability of the extraction: First, we base our extraction on defined criteria for open data [14]. Second, the extraction workflow itself is standardized, and provides guidance within the Numbat tool. Third, extraction was performed by two raters in cases of doubt. The high reliability of the outcome is supported not just by the high IRR, but also by two further observations: First, in the meta-research study by Haven et al. [35], manual confirmation of open data status substantially increased the correlation between the open data status ascribed to supervisor and supervisee articles. Second, we observe only a low one-digit number of open data articles which we missed in the screening, according to the post-hoc registration available to Charité researchers. Although surely some researchers are not aware of the fund or fail to report their missed datasets, this nevertheless indicates an overall low number of clearly open datasets missed in the screening.

The screening workflow presented here has important limitations to consider when applying it, especially in other contexts. We developed it in the context of biomedical research, and it has been validated for biomedical research articles. One consequence of this might be a better detection of certain data types compared to others. Also, due to the nature of our procedure, we miss datasets not related to peer-reviewed research articles (stand-alone datasets or those referenced in preprints). Furthermore, we check only one dataset per repository under the assumption that the openness status of all datasets in a given repository will be similar. This is in line with individual observations, but we did not confirm it quantitatively. Lastly, we include an extraction stream which detects restricted-access datasets. Although we attempt to base this assessment on a solid definition as well [14], we have to acknowledge that for restricted datasets important aspects cannot be checked, and indeed, the decision as such to include restricted-access datasets might not be suitable for other settings where dataset availability is screened.

## Outlook

The open data detection workflow presented here could be applied in the future by institutions or funders interested in data publishing practices or compliance, respectively. In addition, it can be used in meta-research studies. There are several ways in which it could be developed to make it less resource-intensive, and more agile, versatile, and/or reliable.

First, the open data criteria can be made stricter, especially by requiring deposit with either a permanent identifier or an accession code. This would exclude some cases including data on Github, which are particularly time-consuming and difficult to assess (and particularly low in

permanency and reusability). Indeed, we plan to adjust our own criteria along these lines in the future.

Second, the ODDPub tool can be improved in several ways, some of which are already being implemented. This includes a larger set of pre-defined repositories, as well as implementation of aforementioned requirements regarding identifiers. Also, it might be possible to improve the screening by processing of article XMLs rather than PDFs, insofar these are available. Yet another important potential improvement could be the automated detection of data availability statements and, where such a statement is present, ignoring in the automated screening step all remaining text. Please note, however, that the data availability statement can point to other sections of the article for further information on available datasets. Thus, if manual screening were also restricted to the availability statement alone, this could lead to missing references to datasets.

Third, other tools for detecting open datasets have been developed over the last years. The DataSeer tool [42], which has been used to create the PLOS dataset [39], is based on a machine-learning approach and detects several other pieces of dataset-related information. Importantly, this also includes whether a certain article utilized data, allowing to assess the population against which open data rates are calculated. DataStet [43] is another screening tool with additional functionalities, which is derived from DataSeer. Both tools have the potential to improve the detection of open datasets, if combined with ODDPub. Potentially, they could also replace ODDPub altogether, in which case a manual screening step using Numbat would still be possible and useful for our application. Possibly the import into Numbat (or the export from other tools) might need to be adjusted in this case. DataSeer and DataStet have, however, been developed with fully automated screening in mind, and for many purposes these more scalable automated assessments can be sufficient.

Fourth, for certain repositories or data types, shortcuts or predefined templates could be introduced in Numbat. For example, for 2021, over 1/5 of all datasets were deposited in Gene Expression Omnibus. Given the standardized deposition in such repositories, it would be possible to pre-fill most fields for a certain repository with very low probability of decreasing data quality. With four additional disciplinary repositories accounting for an additional nearly 20% of all datasets, this could considerably reduce extraction time. Of course, the extraction could also be adjusted to be faster in several other ways, e.g. by not extracting the license, which is an information we do not currently use (but could in the future, see below).

Fifth, and last, it would be possible to follow more closely the FAIR principles. This would require higher standards, or deviate from the dichotomous yes/no assessment which we currently apply. The current extraction protocol considers some findability and accessibility principles, but hardly considers interoperability and reusability. Especially in cases where a relatively narrow field of research is being surveyed, it would be conceivable to include discipline-specific criteria as e.g. the presence of codebooks, the use of certain ontologies or reporting standards, or the use of certain data types, beyond what we have implemented. In addition, generic criteria on reusability like linkage from the dataset back to the article and the presence of a license can also be implemented.

In sum, we present a workflow for extracting, in a transparent and reproducible way, the information, whether for a research article underlying data have been shared. We hope that this will be used and further developed by others, and will contribute to the detection and promotion of data sharing in different contexts.

## Supporting information

**S1 File. Number of datasets per repository for manually confirmed open data cases in 2021.** Repositories with less than three datasets are not displayed.
(PDF)

**S1 Fig. Flowchart of screening steps to determine the data availability status of articles published in 2020 by researchers from the Charité –Universitätsmedizin Berlin.** Numbers in beige boxes indicate the number of articles screened at the respective stage which complied with the criterion in question. Please note that unlike in Fig 1 und S2 Fig, the numbers refer to an earlier version of the extraction workflow in which we extracted only one dataset per article. Thus, the number of datasets and of articles is identical in this case.
(TIFF)

**S2 Fig. Flowchart of screening steps to determine the data availability status of articles published by researchers at Dutch university medical centers, as screened for the study of Haven et al.** [35]**.** Numbers in beige boxes indicate the number of articles screened at the respective stage which complied with the criterion in question.
(TIFF)

## Acknowledgments

We thank Susan Abunijela, Tamarinde Haven, and Nicole Hildebrand for providing the Dutch UMC dataset. We also thank Jan Taubitz for contributing to the screening of the QUEST dataset, as well as Angela Ariza de Schellenberger, Delwen Franzen, and Martin Holst for giving us feedback to an earlier version of the Numbat extraction workflow. Further, we are thankful to Miriam Kip and Nico Riedel for participating in developing the open data criteria, as well as Nico Riedel for the development of ODDPub and Benjamin Carlisle for the development of Numbat, as well as subsequent user support.

## Author Contributions

**Conceptualization:** Evgeny Bobrov.

**Data curation:** Anastasiia Iarkaeva.

**Formal analysis:** Anastasiia Iarkaeva, Evgeny Bobrov.

**Funding acquisition:** Evgeny Bobrov.

**Investigation:** Anastasiia Iarkaeva, Evgeny Bobrov.

**Methodology:** Anastasiia Iarkaeva, Evgeny Bobrov.

**Project administration:** Evgeny Bobrov.

**Resources:** Vladislav Nachev.

**Software:** Anastasiia Iarkaeva.

**Supervision:** Evgeny Bobrov.

**Validation:** Anastasiia Iarkaeva, Vladislav Nachev, Evgeny Bobrov.

**Visualization:** Anastasiia Iarkaeva.

**Writing – original draft:** Evgeny Bobrov.

**Writing – review & editing:** Anastasiia Iarkaeva, Vladislav Nachev, Evgeny Bobrov.

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
