## [Decision Letter · Decision Letter 0]

26 Dec 2023

PONE-D-23-34484Workflow for detecting biomedical articles with underlying open and restricted-access datasetsPLOS ONE

Dear Dr. Bobrov,

Thank you for submitting your manuscript to PLOS ONE. After careful consideration, we feel that it has merit but does not fully meet PLOS ONE’s publication criteria as it currently stands. Therefore, we invite you to submit a revised version of the manuscript that addresses the points raised during the review process.

We look forward to receiving your revised manuscript.

Kind regards,

Sergio Consoli

Academic Editor

PLOS ONE

Journal Requirements:

Additional Editor Comments:

Please take particular care to the comments raised by all the reviewers (especially R3).

The technical contribution looks not thorough enough. The justification and motivations, along with the objectives of the work should be better elaborated and clearly articulated.

The methodology would require further, deeper investigation. Explain better and motivate the employed dataset and features. The reported computational evaluation and discussion show to be not robust enough, it should be articulated more.

Unfortunately the manuscript fails to meet the PLOS ONE publication criteria in its current form. 

Reviewers' comments:

Reviewer's Responses to Questions

**Comments to the Author**

1. Is the manuscript technically sound, and do the data support the conclusions?

Reviewer #1: Yes

Reviewer #2: Yes

Reviewer #3: Partly

2. Has the statistical analysis been performed appropriately and rigorously? 

Reviewer #1: Yes

Reviewer #2: Yes

Reviewer #3: Yes

3. Have the authors made all data underlying the findings in their manuscript fully available?

Reviewer #1: Yes

Reviewer #2: Yes

Reviewer #3: Yes

4. Is the manuscript presented in an intelligible fashion and written in standard English?

Reviewer #1: Yes

Reviewer #2: Yes

Reviewer #3: Yes

5. Review Comments to the Author

Reviewer #1: The authors do not mention the journals from which they extract the DAS, I would recommend that they do so, to see which journals use this statement, as well as the repositories they have used to verify the presence or not of the data.

On page 18 it says Haven et al, but they do not add the year. On the reference list the year is 2022, but throughout the text it says 2023, is there a missing reference or is it a mistake?

Reviewer #2: The topic of the article is very interesting, important and applicable in current academic system.

The methodology workflow and the results are described clearly, systematically, in detail, and the research seems reproducible. However, since everything else is described in such detail, I suggest you add a few sentences about the protocol for the manual confirmation step. For example, you say “we checked, amongst other criteria, whether the dataset could be found” - how was this checked - using what software, which search terms? Are the raters’ physicians or information specialists? This could affect both the results and the time needed for rating...

I tried to access the data using DOI link provided in Data availability statement, but did not gain access. Will this be open once the article is published?

I have two additional questions that don’t need to be answered in this article, but can serve as “food for taught”:

- Have you considered using other tools which you mention in the discussion on the same set of publications? From the introductory part, it's clear that the results would not be the same, because the methodology and definitions (hence the goals) are different, but it would still be interesting to compare the results and see if your tools and protocols missed anything.

- It would be interesting to compare presence of dataset statements, the actual datasets availability and their accessibility between OA articles and paywalled articles.

Reviewer #3: The article presents a workflow to identify dataset mentions in scholarly articles that are  publicly accessible. To this end, the authors employ existing (or elsewhere introduced) software packages and review the output manually. In particular, ODDPub (developed in the direct environment of the authors) is employed to identify dataset mentions from fulltext pdfs. Numbat (a tool to support systematic reviews) is used then to review the positive decisions of ODDPub. The authors estimate the IRR between 2 and 3 raters on sets of 100 resp. 20 publication and estimate a high reliability. The authors conclude that this manual curation process can be used to estimate the amount of mentions to publicly accessible datasets, but also that the set of publications should not be larger than the sample size used in the paper (~6000). 

I appreciate the idea of identifying dataset mentions automatically to provide extra funding for "Open Scientists". However, I see some problems with the work at hand that should be handled before acceptance.

- From the original work on ODDPub the authors know that the Identification performance is very high. However, the authors note that there are some differences between the evaluation set and the set of articles at hand. I would like to see a more detailed evaluation including articles (at least a sub-sample) that have been sorted out by ODDPub. 

- For an exact definition of what the authors consider as open data, they refer to another publication. As the article should be self-contained, I would like to see the definition in the paper (at least the most interesting parts).

- The article states "... a rater could miss a specific dataset, in which case IRR was calculated only on those datasets extracted by all raters". If I understand this correctly, this introduces a bias in the evaluation, as datasets that where not identified by a rater a skipped during evaluation. When applying the final workflow with only one rater, those datasets would be missing entirely. 

- When it comes to the interpretation of the IRR, the authors argue that maximizing the IRR was not focus of the study but the generation of reliable data. This should be the default case to estimate the quality. Later, the authors further argue that using the class category "unsure" less liberally, would further increase the IRR. I don't understand this argumentation, as the final goal is to get all important information from the articles and not optimizing the IRR. The analysis of the IRR reads a bit like "we could have cheated, but we didn't"

- In the interpretation of the results, it is argued that the high reliability is not only supported by the high IRR, but by an increased correlation of another study that employed the same protocol. In general, correlation could also be increased by additional errors, thus the authors should elaborate on this and state in how far this supports their results.

- The results of Krippendorffs alpha is interpreted by the approach of Landis&Koch 1977, who base their work on Cohen's kappa. Is it reliable to do this generalization?

- When evaluating the duration of the manual screening, the authors estimate 4.5 minutes per article. This shows that the process does not scale well. Further, the authors state that actual time to spend is much higher due to unsure cases. I'm puzzled by this statement. What is the purpose of the estimation, when the authors do not believe the result. Also, while the authors state that 4 minutes seems to be low, I have the feeling that it is rather high just to check if an identified dataset is publicly available or not. Please elaborate!

- The novel parts of the process are mainly based on manual work. It would be nice if the authors outline in how far the entire process could be automated to eventually apply the incentivization on a regular basis.

 - Figure 1 is missing the articles that could actually be retrieved due to closed or restricted access. Further, in order to make a statement about potential application in other setting it might be interesting to get an idea of the amount of open access articles in comparison with others.

Beside the issues listed above, I see a couple of technical issues that could easily be fixed. (The order does not reflect importance):

- In the abstract, the authors state that an IRR of >0.8 was estimated, but there is no information about what kind of measure was used for estimation.

- The KrippAlpha function from the DescTools package was used. The function was cited, but not the packages. I suggest to cite the package directly.

- The authors cite all software that was used in the study and use links to archived versions. However, neither a software version, nor the data of access was provided. While it would of course be possible to identify the software via software heritage, software citations should be accompanied by version (or dates).

- It might be good citation style to cite websites and software without date, I have the feeling that it would help the reader to add a date.

- The link to the dataset published with the study is wrong, as it points to some sharepoint folder

- When estimating the amount of open data the article states "... that 7.9% of articles had underlying openly available data. In addition, 1.05% of articles had shared restricted-access datasets, resulting in an overall 8.4% of articles for which at least one dataset was available". I would guess that the final value should be 8.95%. Please correct!

- There are some issues with the bibliography:

 - Bobrov et al (2023) is regularly cited but not provided in the list of references

- Haven et al., 2023 is cited but in the list of references provided as "Haven, T. L., Abunijela, S., & Hildebrand, N. (2022, September 15). Biomedical supervisors’ role modeling of responsible  research practices: a cross-sectional study."

 - Serghiou et al. (2021) is missing in the references

6. PLOS authors have the option to publish the peer review history of their article (what does this mean?). If published, this will include your full peer review and any attached files.

Reviewer #1: **Yes: **Remedios Melero

Reviewer #2: No

Reviewer #3: **Yes: **Frank Krüger

---

## [Author Response · Author response to Decision Letter 0]

4 Mar 2024

*please be aware that the formatting is lost here, and it will probably be much easier to read our reply in the PDF file*

As additional information to all reviewers, apart from addressing their comments:

Importantly, some numbers in the manuscript which relate to the Charité 2021 screening have slightly changed, as we corrected the number of article which were screened with ODDPub overall. We had reported this number to be 5044, but indeed it is 5119. This does not change any conclusions, and largely led to changes in the first decimal of percentages. The difference of 75 articles is due to a discrepancy between our dashboard and the article set reported here. We had based our calculations on the number in the dashboard, but closer scrutiny showed that the sample is not fully identical, due to different time points at which the samples were drawn from publication databases.

Additional Editor Comments:

Reviewer #1: The authors do not mention the journals from which they extract the DAS, I would recommend that they do so, to see which journals use this statement, as well as the repositories they have used to verify the presence or not of the data.

With regard to repositories, we have added an analysis of the repositories in which datasets underlying publications from 2021 have been shared. The table has been added as S1 Table. Three datasets were excluded to avoid double-dipping, as each of them had been underlying the analysis in two articles published in the same validation period. Importantly, we only extracted one dataset per article and repository. Thus, the numbers in the table do not correspond to the overall number of datasets deposited. To extract this would not only have been much more time-consuming, but would require a more detailed operationalization of what a dataset is. Despite this limitation, the numbers indicate which repositories are commonly used in biomedical research, and indicate a dominance of repositories for genetic data, as already visualized in our FAIR Data Dashboard.

With regard to journals, the information which journal provided DAS and which didn’t is not available to us. We screened article whole texts for statements indicative of data availability, and did not restrict ourselves to the DAS. We also did not specifically document whether a DAS was present. This is in line with our statement in the article that extracting DAS could be a future improvement to the algorithm, and indeed, the newest version of the algorithms approaches that question. 

On page 18 it says Haven et al, but they do not add the year. On the reference list the year is 2022, but throughout the text it says 2023, is there a missing reference or is it a mistake?

Thank you for pointing this out. There was a mistake as we still listed the preprint from 2022 in the reference list, but in the meanwhile a reviewed article was published in 2023. The reference is now to the 2023 reviewed article.

Reviewer #2: The topic of the article is very interesting, important and applicable in current academic system.

The methodology workflow and the results are described clearly, systematically, in detail, and the research seems reproducible. However, since everything else is described in such detail, I suggest you add a few sentences about the protocol for the manual confirmation step. For example, you say “we checked, amongst other criteria, whether the dataset could be found” - how was this checked - using what software, which search terms? Are the raters’ physicians or information specialists? This could affect both the results and the time needed for rating...

We have provided a complete list of the individual steps in the extraction, and we refer to the figures where this is shown. We have also added some information on the professional background of the raters. Details on the manual checking are described in an open protocol (Iarkaeva et al., 2022), to which we already refer in this paragraph, and thus, to keep it readable, we have kept the additional text short.

I tried to access the data using DOI link provided in Data availability statement, but did not gain access. Will this be open once the article is published?

Having the data available online is a matter of course for us, and the dataset was already online at the time of submission. Unfortunately, while the DOI was correct, something went wrong with the link, thank you for pointing this out.

I have two additional questions that don’t need to be answered in this article, but can serve as “food for taught”:

- Have you considered using other tools which you mention in the discussion on the same set of publications? From the introductory part, it's clear that the results would not be the same, because the methodology and definitions (hence the goals) are different, but it would still be interesting to compare the results and see if your tools and protocols missed anything.

This is also of high interest to us, and we would definitely like to conduct such an analysis. We are particularly interested in the tool DataStet, used to create both the French Open Science Monitor and the PLOS open science indicators dataset. The first step would probably be to apply DataStet to Charité publications and compare the outcome with our results. Indeed, we consider doing so this year, but this would be out of scope of the present article.

- It would be interesting to compare presence of dataset statements, the actual datasets availability and their accessibility between OA articles and paywalled articles.

We appreciate the suggestion to compare the rates of data sharing for open access and paid access articles, and have added the analysis to the article. As to be expected, sharing data is more common for open access articles, which have a 9.5% probability of having manually confirmed open or restricted-access data, as opposed to 4.8% for paywalled articles. Importanly, this analysis only applies to the articles which we downloaded and screened with ODDPub at the time of validation (i.e., in 2022). For all articles of the Charité, this can only serve as an approximation, as we screened a large majority of Charité articles (80.0%), but not all. Articles were not screened for one or more of the following reasons (as also listed below in an answer to reviewer #3): (i) lacking institutional access to certain paid-access journals, (ii) failure to automatedly download open access articles for technical reasons, often due to barriers against automated download placed by the journals, and (iii) deliberate exclusion of some article types as e.g. obituaries and corrigenda to reduce the computational workload. While excluded article types should indeed not be part of this analysis, the other two cases should be in the sample to make a definitive statement about the share of available data by article type. 

With respect to the other analyses suggested, we are not fully sure what is meant. If “presence of dataset statements” refers to statements as detected by ODDPub, this would only add the false positive cases which we manually removed. And if “accessibility” refers to one of the steps during the checking process, the same question could then be asked about all other extraction steps as well, e.g. “findability” and “deposition in a repository”. As this is mentioned as “food for thought” we assume that this is not an issue that needs to be resolved, but if it is a point important to address, we would ask the reviewer to provide more detail on this suggestion.

Reviewer #3: The article presents a workflow to identify dataset mentions in scholarly articles that are publicly accessible. To this end, the authors employ existing (or elsewhere introduced) software packages and review the output manually. In particular, ODDPub (developed in the direct environment of the authors) is employed to identify dataset mentions from fulltext pdfs. Numbat (a tool to support systematic reviews) is used then to review the positive decisions of ODDPub. The authors estimate the IRR between 2 and 3 raters on sets of 100 resp. 20 publication and estimate a high reliability. The authors conclude that this manual curation process can be used to estimate the amount of mentions to publicly accessible datasets, but also that the set of publications should not be larger than the sample size used in the paper (~6000). 

I appreciate the idea of identifying dataset mentions automatically to provide extra funding for "Open Scientists". However, I see some problems with the work at hand that should be handled before acceptance.

- From the original work on ODDPub the authors know that the Identification performance is very high. However, the authors note that there are some differences between the evaluation set and the set of articles at hand. I would like to see a more detailed evaluation including articles (at least a sub-sample) that have been sorted out by ODDPub. 

We agree that since some time has passed and we introduced some changes, an extensive validation including a screen of a set of articles not flagged as open data articles by ODDPub would be ideal. However, given that the changes we introduced are relatively minor, we think that it is justified to still refer primarily to the validation performed for the article on ODDPub (Riedel et al., 2020). At the same time, to assess whether the former validation at least still appears plausible, we have screened a small set of 100 articles manually using the same screening method we had used to validate the ODDPub algorithm for aforementioned publication. To do so, we searched in the fulltext for keywords and word roots like “data”, “dataset”, “access*”, “availab*” etc. We found that ODDPub missed one restricted-access case. This suggests a false negative rate of 1%, which is lower than the 3.4% reported in Riedel et al. However, 17 out of 24 cases missed there were supplemental materials, which we now exclude. Thus, the numbers are not directly comparable, but excluding supplemental materials would have led to a rate of 1.0% in the Riedel et al. sample, which would be well in line with the value observed now. Of course, it must be acknowledged that 100 articles are a small sample for such a validation, given the low probability of false negatives, and thus the value of 1% we find now should only serve as supportive evidence in light with our much more extensive previous evaluation of 792 articles. We have added this in the methods section. 

- For an exact definition of what the authors consider as open data, they refer to another publication. As the article should be self-contained, I would like to see the definition in the paper (at least the most interesting parts).

We agree that an overview of the criteria for available datasets is necessary, and we have added that to the manuscript. More detail than what we added would seem to us beyond the scope of this article, but we state more clearly now why the definition is complex, and how it is covered in Bobrov et al., 2023.

- The article states "... a rater could miss a specific dataset, in which case IRR was calculated only on those datasets extracted by all raters". If I understand this correctly, this introduces a bias in the evaluation, as datasets that where not identified by a rater a skipped during evaluation. When applying the final workflow with only one rater, those datasets would be missing entirely. 

 Old IRR Nr. of datasets (old) New IRR Nr. of datasets (new)

3 raters (dataset level) 0.762 26 0.793 31

2 raters (dataset level) 0.819 146 0.81 169

3 raters (article level) 0.826 20 0.826 20

2 raters (articles level) 0.901 100 0.901 100

We agree that this for an overall assessment of the reliability of results, the measure suggested by the reviewer is better. We have thus additionally calculated the IRR as suggested. The numbers in the table above indicate that if we explicitly consider datasets only detected by some raters, but not others, this does not substantially change the IRR. On the contrary, for three raters the IRR value even increased (see (iv) for more details). For two raters and a larger set of extractions, the IRR decreased very slightly from 0.819 to 0.810. We have added information on this analysis to the methods, and now report numbers on the “new” way of analyzing dataset-level IRR in the results.

It is an important point whether there is the possibility for those cases where only one rater screened an article to have misclassified it altogether, or what the probability of such an event is. In addition to the change in IRR after recalculation being only minor, several other lines of reasoning support that the probability of completely misclassifying an article (i.e., classifying it as “open data” where this is not the case, and vice versa) is very low:

(i) In the 100 articles extracted independently by two raters, it was never the case that one rater detected an available dataset, the other did not, and the final assessment was that there was indeed an available dataset; this case would have allowed for a false negative, but it never occurred; we have added this information to the discussion section of the article

(ii) In the same 100 article, the reverse case occurred only once, where one rater detected an available dataset, the other rater did not detect it, and the overall assessment was that the dataset was not an available dataset by our definition; this case could have allowed for a false positive

(iii) Importantly, the case described under (ii) was a rare case of data underlying a systematic review, and when consulting an independent expert on systematic review, the initial assessment of the case did not agree with our final assessment; this indicates that the case is a borderline case, while also showing the limits in precision, especially if the criteria described in Bobrov et al. (2023) are not applied meticulously

(iv) For the 20 articles screened by three raters, three datasets were not detected by one of the raters, and one dataset was not detected by two out of three raters. All of these datasets were so-called “source data”. These are supplementary data and were ultimately never considered open data, but due to our decision to exclude supplementary data, some raters did not even begin to extract them. While this indicates degrees of flexibility in the exact procedure, this has no impact on the overall assessment, and indicates that the cases which differed between raters can typically be easily avoided.

(v) Lastly, the probability of missing a dataset can be expected to be inversely correlated to the number of datasets extracted per article. If ODDPub flagged an article, the rater can be expected to look for an available dataset very thoroughly. However, ODDPub does not indicate the number of available datasets, and it is very probably easier to miss additional datasets. 

- When it comes to the interpretation of the IRR, the authors argue that maximizing the IRR was not focus of the study but the generation of reliable data. This should be the default case to estimate the quality. Later, the authors further argue that using the class category "unsure" less liberally, would further increase the IRR. I don't understand this argumentation, as the final goal is to get all important information from the articles and not optimizing the IRR. The analysis of the IRR reads a bit like "we could have cheated, but we didn't"

Even though this is surely not what we wanted to communicate, we understand why our point did not come across well. Definitely improving the information was and must be the goal. However, there are different levels of information at play, which are potentially in conflict: the individual assessment level, and the overall assessment level. We wanted to point out here that we accepted a lower precision of the individual decision to channel these cases into the reconciliation procedure and thus ultimately increase the precision of the overall decision. We have adjusted the manuscript accordingly.

- In the interpretation of the results, it is argued that the high reliability is not only supported by the high IRR, but by an increased correlation of another study that employed the same protocol. In general, correlation could also be increased by additional errors, thus the authors

---

## [Decision Letter · Decision Letter 1]

12 Apr 2024

Workflow for detecting biomedical articles with underlying open and restricted-access datasets

PONE-D-23-34484R1

Dear Dr. Bobrov,

We’re pleased to inform you that your manuscript has been judged scientifically suitable for publication and will be formally accepted for publication once it meets all outstanding technical requirements.

Kind regards,

Sergio Consoli

Academic Editor

PLOS ONE

Additional Editor Comments (optional):

Reviewers' comments:

Reviewer's Responses to Questions

**Comments to the Author**

1. If the authors have adequately addressed your comments raised in a previous round of review and you feel that this manuscript is now acceptable for publication, you may indicate that here to bypass the “Comments to the Author” section, enter your conflict of interest statement in the “Confidential to Editor” section, and submit your "Accept" recommendation.

Reviewer #1: All comments have been addressed

Reviewer #2: All comments have been addressed

Reviewer #3: All comments have been addressed

2. Is the manuscript technically sound, and do the data support the conclusions?

Reviewer #1: Yes

Reviewer #2: (No Response)

Reviewer #3: Yes

3. Has the statistical analysis been performed appropriately and rigorously? 

Reviewer #1: Yes

Reviewer #2: (No Response)

Reviewer #3: Yes

4. Have the authors made all data underlying the findings in their manuscript fully available?

Reviewer #1: Yes

Reviewer #2: (No Response)

Reviewer #3: Yes

5. Is the manuscript presented in an intelligible fashion and written in standard English?

Reviewer #1: Yes

Reviewer #2: (No Response)

Reviewer #3: Yes

6. Review Comments to the Author

Reviewer #1: I have not any further questions, I think authors have responded all reviewers queries and it deserves to be published

Reviewer #2: (No Response)

Reviewer #3: I appreciate the detailed answer and the explanations provided by the authors. I don't have any additional comments.

7. PLOS authors have the option to publish the peer review history of their article (what does this mean?). If published, this will include your full peer review and any attached files.

Reviewer #1: **Yes: **Remedios Melero

Reviewer #2: No

Reviewer #3: **Yes: **Frank Krüger

---

## [Editor Report · Acceptance letter]

26 Apr 2024

PONE-D-23-34484R1 

PLOS ONE

Dear Dr. Bobrov, 

I'm pleased to inform you that your manuscript has been deemed suitable for publication in PLOS ONE. Congratulations! Your manuscript is now being handed over to our production team.

Kind regards, 

on behalf of

Dr. Sergio Consoli 

Academic Editor

PLOS ONE